# Federated learning framework based on trimmed mean aggregation rules

## Abstract

This paper studies the problem of information security in the distributed learning framework. In particular, we consider the clients will always be attacked by Byzantine nodes and poisoning in the federated learning. Typically, aggregation rules are utilized to protect the model from the attacks in federated learning. However, classical aggregation methods such as $\mathrm{Krum}(\cdot)$ and $\mathrm{Mean}(\cdot)$ are not capable enough to deal with Byzantine attacks in which general deviations and multiple clients are attacked at the same time. We propose new aggregation rules, $\mathrm{Tmean}(\cdot)$, to the federated learning algorithm, and propose a federated learning framework based on Byzantine resilient aggregation algorithm. Our $\mathrm{Tmean}(\cdot)$ rules are derived from $\mathrm{Mean}(\cdot)$ by appropriately trimming some of the values before averaging them. Theoretically, we provide theoretical analysis and understanding of $\mathrm{Tmean}(\cdot)$. Extensive experiments validate the effectiveness of our approaches.

## 1 Introduction

As one special case of distributed machine learning, federated learning (FL) draws increasing research attention recently. FL has become one promising approach to enable clients collaboratively to learn a shared model with the decentralized and private data on each client node. Thus it is of central importance to keep the node information secured in FL. Unfortunately, FL is very vulnerable to software/hardware errors and adversarial attacks; especially Byzantine attack from distributed systems has arose to be a key node attack sample for federated learning.

To ensure the FL model resistance to Byzantine attack, research focuses are made on *how to introduce aggregation rules into the gradient information iteration process*, and *how to ensure that this aggregation rule can make the data robust*. In particular, classical approaches to avoid the Byzantine failures would employ the state machine replication strategy (Alistarh et al., 2018), which can be roughly categorized into two ways in distributed machine learning: (1) the processes agree on a sample of data based on which the clients update their local parameter vectors; (2) the clients agree on how the parameter vector should be updated (Blanchard et al., 2017). The former ones demand transmitting data samples to each individual node, resulting in high costs. The latter ways are not reliable neither, as we cannot detect whether clients are trustworthy or not, from the mixed Byzantine vectors. The attacker can know any information about the process of FL, and can use any vectors to initiate the attack during the node's information transmission (Yin et al., 2018). More specifically, the data between machines can be replaced by any value. This problem is not fully addressed in previous works.

When facing the attacks from Byzantine nodes, the FL models have to rely on robust aggregation rules to minimize the influence of the Byzantine attack on the data model (Bottou, 2010). For instance, $\mathrm{Krum}(\cdot)$ is a strong aggregation rule designed to identify an honest node such that it can effectively prevent Byzantine attacks in most cases (Blanchard et al., 2017). However, if the Byzantine node attack is changed from the original single miner node attack to multiple server attacks, $\mathrm{Krum}(\cdot)$ is not able to ensure the learning process robustness to noisy data. As another class of aggregation rules, simple mean-based aggregation rules can maintain learning robustness if it is not attacked by Byzantine nodes. However, the Byzantine attack will make the update direction of node gradient information largely deviate from the original function by using simple mean-based aggregation rules. Some variants of mean-based aggregation rules, such as *geometric median* (Blanchard

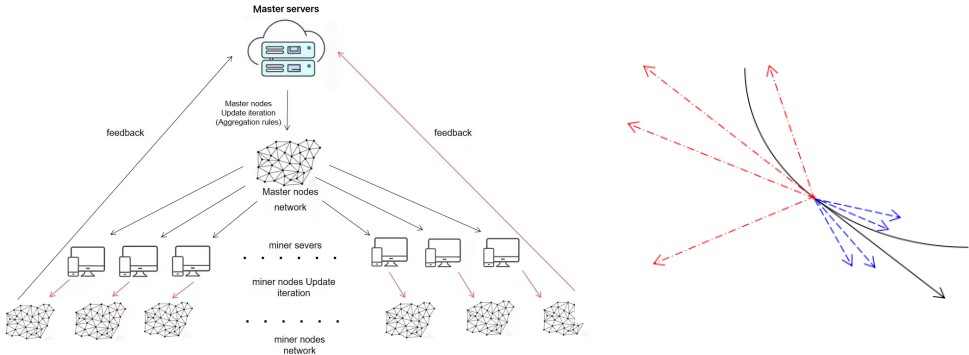

(a) Federated learning gradient update iteration     (b) SGD gradient update

Figure 1: (a) is the procession of FL gradient updating. It consists of four parts: master server, master server distributed network, miner servers and miner network. (b) represents the schematic diagram of the gradient update direction: blue dashed lines represent the estimated gradients of miner nodes; red dashed lines represent the Byzantine update gradients under the attack of the Byzantine nodes; black solid line represents the ideal gradient.

et al., 2017), *marginal median* (Alistarh et al., 2018), are the classical methods to solve the Byzantine node attack problem. But they are not much more robust than simple mean-based aggregation rules in the case of some large deviation Byzantine attacks. The main difficulty that prevents mean-based aggregation rules from robust is the unstable distribution of data (Jin et al., 2020). We find that the difficulty can be tackled, if the data is averaged by trimming a part of the data and then imported into the aggregation rules. This motivates our work in this paper.

Most of FL approaches are built upon the Stochastic Gradient Descent (SGD) algorithm (Castro et al., 1999) or its variants, and the statistical properties of SGD such as convergence are well developed. Our approach also employs the SGD; and the typical iterative process of SGD algorithm in distributed system is represented by Figure 1. First, a client, known as a miner node, estimates the gradient of the node, makes an estimate of the deviation between the estimated information and the ideal gradient information (El-Mhamdi et al., 2020), then passes this information to the server node in the network, and finally update the gradient there. So the miner node network transmits the information to the master server, and the master server then passes the gradient update information through a series of aggregation operations. When the aggregation conditions are met, the server transmits the information to the distribution network, and then transmitted to the miner nodes. This is a cyclic process, such that the gradient information is continuously updated in the entire network (Li et al., 2014).

In this paper, we mainly propose new aggregation rules, $\mathrm{Tmean}(\cdot)$, by trimming part of the data before the average operation. We provide theoretical understandings of our aggregation rules, especially, why they are robust to the Byzantine attack. Through attack experiments and mathematical proofs, we have concluded that appropriately trimming the original data and averaging can make the model more robust from the decentralized data.

Specifically, in section 3 we introduce the federated learning Byzantine distributed system model, briefly describes the working principle of SGD, and summarizes the update iteration rules based on aggregation rules. Then, we present the concept of Byzantine resilience, and the conditions to satisfy the aggregation rules of Byzantine resilience. We provide concept of trimmed mean and the rigorous theoretical proof and understanding of $\mathrm{Tmean}(\cdot)$ based aggregation rules in section 4. Then, we prove the convergence of the proposed federated learning aggregation rules in section 5. In section 6, we conduct experiments by Gaussian attack, omniscient attack and multiple servers attack. Under these attacks, $\mathrm{Tmean}(\cdot)$-based FL aggregation rules can still maintain robustness. These experiments thus validate the effectiveness of our approaches.

**Contributions:** (1) We present new aggregation rules, $\mathrm{Tmean}(\cdot)$, to the Byzantine resilient federated learning algorithm, and propose federated learning frameworks based on aggregation algorithm. Our proposed approaches are shown to be robust to Byzantine attacks.

(2) We provide rigorous theoretical proof and understanding of our approaches and aggregation rules. To the best of our knowledge, these theorems and theoretical understandings are for the first time contributed to the community. Critically, we make convergence certificates and prove that $\mathrm{Tmean}(\cdot)$ can converge in the general convex optimization setting.

(3) Empirically, we demonstrate that the effectiveness of our approaches can make the FL model robust to Byzantine attack.

## 2 RELATED WORK

**Federated Learning.** Federated learning has become a prominent distributed learning paradigm. In (Jin et al., 2020), the authors proposed Stochastic-Sign SGD, a parameter estimation method with convergence guarantees. It uses a gradient compressor based on random symbols to unify that the gradient is updated in the framework. The FedAvg algorithm was first proposed by (Konečný et al., 2016), and many scholars subsequently improved it based on this algorithm. (Karimireddy et al., 2020) pointed out that FedAvg can be improved by adding one additional parameters control variables to correct the client drift, and proved that FedAvg may be seriously affected by the gradient difference of different clients, and may even be slower than SGD.

**Byzantine attack in FL.** To make the secure transmission of the master server node, the FL models have to deal with the Byzantine attack. (Yin et al., 2018) showed that certain aggregation rules in the master server node can ensure the robustness of the data. (Blanchard et al., 2017) pointed out that $\mathrm{Krum}(\cdot)$ can be used to make the robustness of data by computing the local sum of squared Euclidean distance to the other candidates, and outputting the one with minimal sum. When the gradient is updated to the saddle point, there is no guarantee that the SGD algorithm converges to the global optimum. Some scholars have proposed ByzantinePGD (Yin et al., 2019), which can escape saddle points and false local minimums, and can converge to an approximate true local minimum with low iteration complexity. Later, in the strong anti-Byzantine model, some scholars carried out poisoning attacks on Byzantine robust data, from the original data collection process to the subsequent information exchange process, and at the same time, these attacks were defended accordingly, in (Zhao et al., 2021). The research on the Byzantine structure model of federated learning has been expanded once again (Zhao et al., 2021).

**Attack and Defence.** In the framework of federated learning, miner nodes are usually attacked, such as Byzantine node attacks, poisoning attacks (Zhao et al., 2021), gradient leakage (Wei et al., 2020), etc. There exist various defense methods also, such as robust aggregation, secure aggregation, encryption, etc. Our paper mainly studies Byzantine node attacks. Byzantine node attacks can usually be summarized into two types: (1) To update gradient information of nodes, nodes are replaced by Byzantine nodes, and normal worker nodes cannot make judgments so that the gradient estimates deviate from the actual gradient update direction. (2) During the gradient process, the local node suffering from the interference of the Byzantine node, cannot reach a consensus with the master node, making the entire process unable to proceed normally (Lamport et al., 2019). $\mathrm{Krum}(\cdot)$ is a popular aggregation rule to deal with these node attacks. We shall propose $\mathrm{Tmean}(\cdot)$ as alternative aggregation rules.

## 3 PRELIMINARY AND PROBLEM SETUP

### 3.1 FEDERATED LEARNING SETTING

Federated learning was first proposed in (Konečný et al., 2016), where the prevalent asynchronous SGD is used to update a global model in a distributed fashion. A pioneering work in this field proposed the currently most widely used algorithm, FedAvg (McMahan et al., 2017), which is also the first synchronous algorithm dedicated to federated setting. Recent studies attempt to expand federated learning with the aim of providing learning in more diverse and practical environments

---

**Server:**
Initialize $x_0 \leftarrow$ `rand()`;
**for** $t = 0, 1, \ldots, T$ **do**

    Broadcast $x^{(t)}$ to all the workers;

    Wait until all the gradients $\tilde{v}_1^{(t)}, \tilde{v}_2^{(t)}, \ldots, \tilde{v}_m^{(t)}$ arrive;

    Compute $G^{(t)} = \text{Aggr}(\tilde{v}_1^{(t)}, \tilde{v}_2^{(t)}, \ldots, \tilde{v}_m^{(t)})$;

    Update the parameter $x^{(t+1)} \leftarrow x^{(t)} - \gamma^{(t)} G^{(t)}$;

**end**

**Worker:**
**for** $t = 0, 1, \ldots, T$ **do**

    Receive $x^{(t)}$ from the server;

    Compute and send the local randomized gradient $v^{(t)} = \nabla F(x^{(t)}, \xi_k)$ to the server;

**end**

**Algorithm 1:** Distributed synchronous SGD with robust aggregation server.

---

such as multi-task learning, generative models, continual learning, and data with noisy labels. However, these algorithms may obtain suboptimal performance when miners participating in FL have non-independently and identical distributions (non-i.i.d.) (Zhao et al., 2018).

While the convergence of FedAvg on such settings was initially shown by experiments in (McMahan et al., 2017), it does not guarantee performance as good as that in an i.i.d. setting. These algorithms pointed out the issue have major limitations, such as privacy violation by partial global sharing of local data (Zhao et al., 2018) or no indication of improvement over baseline algorithms such as FedAvg (Hsieh et al., 2020). Our paper focuses on a general federated setting.

In order to better study the problem of aggregation in federated learning, we consider the following optimization problem:

$$\min_{x \in \mathbb{R}^d} F(x), \tag{1}$$

where $F(x) = \mathsf{E}_{\xi \sim \mathcal{D}}[f(x; \xi)]$ is a smooth convex function, $\xi$ is sampled from some unknown distribution $\mathcal{D}$. Ideally, the problem (1) can be solved by the gradient descent method as

$$x^{(t+1)} \leftarrow x^{(t)} - \gamma^{(t)} \nabla F(x^{(t)}), \tag{2}$$

where $\gamma^{(t)}$ is the learning rate at $t$-th round. We assume that there exists at least one minimizer of $F(x)$, denoted by $x^*$, that satisfies $\nabla F(x^*) = 0$.

The problem (1) is solved in a distributed manner with $m$ miner nodes, and up to $q$ of them may be Byzantine nodes. The detailed algorithm of distributed synchronous SGD with aggregation rule $\text{Aggr}(\cdot)$ is shown in Algorithm 1. In each iteration, each miner samples $n$ i.i.d. data points from the distribution $\mathcal{D}$, and computes the gradient of the local empirical loss. Using a certain aggregation rule $\text{Aggr}(\cdot)$, the server collects and aggregates the gradients sent by the miners, and estimate the gradient through $\nabla F(x^{(t)}) \approx \text{Aggr}(v_1, v_2, \ldots, v_m)$. Without Byzantine failures, the $k$-th worker calculates $v_k^{(t)} \sim G^{(t)}$, where $G^{(t)} = \nabla f(x^{(t)}, \xi)$. When there exist Byzantine faults, $v_k^{(t)}$ can be replaced by any arbitrary value $\tilde{v}_k^{(t)}$, and hence

$$\nabla F(x^{(t)}) \approx \text{Aggr}(\tilde{v}_1, \tilde{v}_2, \ldots, \tilde{v}_m). \tag{3}$$

### 3.2 BYZANTINE RESILIENCE

In the following we introduce the concept of *Byzantine resilience*. Suppose that in a specific iteration, the correct vectors $v_1, v_2, \ldots, v_m$ are i.i.d. samples drawn from the random variable $G = \nabla f(x; \xi_k)$, where $\mathsf{E}[G] = g$ is an unbiased estimator of the gradient based on the current parameter $x$. Thus, $\mathsf{E}[v_k] = \mathsf{E}[G] = g$ for $k \in \{1, 2, \ldots, m\}$. We simplify the notations by omitting the index of iteration $t$. The following Definition 1 defines the concept of $\Delta$-Byzantine resilience. For simplicity, we sometimes use Byzantine resilient in short when the concrete value of $\Delta$ is either clear from the context or not important.

**Definition 1** (Byzantine Resilience). *Suppose that $0 \leq q \leq m$, and $\Delta$ is a given positive number. Let $v_1$, $v_2$, ..., $v_m$ be i.i.d. random vectors in $\mathbb{R}^d$, where $v_k \sim G$ with $\mathsf{E}[G] = g$. Let $\tilde{v}_1$, $\tilde{v}_2$, ..., $\tilde{v}_m$ be* attacked *copies of $v_k$'s, such that at least $m - q$ of which are equal to the corresponding $v_k$'s while the rest are arbitrary vectors in $\mathbb{R}^d$. An aggregation rule* $\mathrm{Aggr}(\cdot)$ *is said to be $\Delta$-Byzantine resilient if*

$$\mathsf{E}\big[\|\mathrm{Aggr}(\tilde{v}_1, \tilde{v}_2, \ldots, \tilde{v}_m) - g\|^2\big] \leq \Delta.$$

## 4 FL FRAMEWORK WITH ROBUST AGGREGATION

Nowadays, the aggregation rules based on the FL framework are mainly based on $\mathrm{Krum}(\cdot)$ and $\mathrm{Mean}(\cdot)$. They are usually used in the defense of Byzantine node attacks. However, the aggregation rules based on $\mathrm{Mean}(\cdot)$ often cannot exhibit strong robustness, if it is affected by Byzantine nodes. The attack makes the gradient estimation direction deviate too far from the actual gradient update direction; and it cannot guarantee the robustness of the data. Thus we shall propose trimmed mean-based aggregation rules to resolve this issue.

### 4.1 KRUM

The Krum rule is a strong aggregation rule that attempts to identify an honest computing node, and discards the data of other computing nodes. The data from the identified honest computing node is used in the next round of algorithm. The selection strategy is to find one whose data is closest to that on other computing nodes. In other words, it computes the local sum of squared Euclidean distances to the other candidates, and outputs the one with minimal sum. A notable property of $\mathrm{Krum}(\cdot)$ is shown in Lemma 1.

**Lemma 1** ((Blanchard et al., 2017)). *Let $v_1$, ..., $v_m \in \mathbb{R}^d$ be any i.i.d. random vectors such that $v_k \sim G$ with $\mathsf{E}[G] = g$ and $\mathsf{E}\big[\|G - g\|^2\big] \leq V$. Let $\tilde{v}_1$, $\tilde{v}_2$, ..., $\tilde{v}_m$ be attacked copies of $v_k$'s, where up to $q$ of them are Byzantine. If $2q + 2 < m$, then* $\mathrm{Krum}(\cdot)$ *is $\Delta_0$-Byzantine resilient, where*

$$\Delta_0 = \left(6m - 6q + \frac{4q(m - q - 2) + 4q^2(m - q - 1)}{m - 2q - 2}\right) V.$$

### 4.2 MEAN-BASED AGGREGATION RULES

An important class of aggregation rules is mean-based aggregation rules. Mathematically, an aggregation rule $\mathrm{Aggr}(\cdot)$ is a *mean-based* one if $\mathrm{Aggr}(v_1, v_2, \ldots, v_m)$ is guaranteed to be a convex combination of $v_1$, $v_2$, ..., $v_m$. Roughly speaking, a mean-based rule produces an averaged value of the input data. For instance, the arithmetic mean $\mathrm{Mean}(v_1, v_2, \ldots, v_m) = \frac{1}{m} \sum_{k=1}^{m} v_k$ is the simplest mean-based aggregation rule. However, simple mean-based aggregation rules are in general *not* robust under Byzantine attacks, in the sense that they do *not* satisfy Byzantine resilience conditions (Yin et al., 2018).

In the following, we show that by carefully handling the data, certain mean-based rules can satisfy Byzantine resilience conditions. We first introduce the concept of *b-trimmed means* based on order statistics as in Definition 2. This concept is closely related to the coordinate-wise trimmed mean defined in (Yin et al., 2018).

**Definition 2.** *Let $u_1$, $u_2$, ..., $u_m$ be scalars whose order statistics is*

$$u_{(1)} \leq u_{(2)} \leq \cdots \leq u_{(m)}.$$

*For an integer $b$ with $0 \leq b < m/2$, the b-trimmed mean of $u_1$, $u_2$, ..., $u_m$ is defined as*

$$\mathrm{Tmean}_b(u_1, u_2, \ldots, u_m) = \frac{1}{m - 2b} \sum_{k=b+1}^{m-b} u_{(k)}.$$

*For d-vectors $v_1$, $v_2$, ..., $v_m \in \mathbb{R}^d$, the b-trimmed mean $\mathrm{Tmean}_b(v_1, v_2, \ldots, v_m) \in \mathbb{R}^d$ is defined in the component-wise sense, i.e.,*

$$\mathrm{Tmean}_b(v_1, v_2, \ldots, v_m) = \sum_{i=1}^{d} e_i \cdot \mathrm{Tmean}_b\big(\langle v_1, e_i \rangle, \langle v_2, e_i \rangle, \ldots, \langle v_m, e_i \rangle\big),$$

*where $e_i$ is the $i$-th column of the $d \times d$ identity matrix.*

Intuitively, aggregation rules using $b$-trimmed means have opportunities to discard abnormal values from the input data. Lemma 2 shows that $b$-trimmed means are usually within a meaningful range under mild assumptions.

**Lemma 2.** *Let $u_1$, $u_2$, ..., $u_m$ be scalars whose order statistics is*

$$u_{(1)} \leq u_{(2)} \leq \cdots \leq u_{(m)},$$

*and $\tilde{u}_1$, $\tilde{u}_2$, ..., $\tilde{u}_m$ be attacked copies of $u_1$, $u_2$, ..., $u_m$ with up to $q$ Byzantine elements. If the order statistics of $\tilde{u}_k$'s is*

$$\tilde{u}_{(1)} \leq \tilde{u}_{(2)} \leq \cdots \leq \tilde{u}_{(m)},$$

*then for $q \leq b < m/2$, we have*

$$u_{(k-b)} \leq u_{(k-q)} \leq \tilde{u}_{(k)} \leq u_{(k+q)} \leq u_{(k+b)}, \qquad (b+1 \leq k \leq m-b).$$

*Proof.* Let $\tilde{u}_{(k_0)}$ be the largest non-Byzantine element in $\{\tilde{u}_{(1)}, \tilde{u}_{(2)} \ldots, \tilde{u}_{(k)}\}$. Since the number of Byzantine elements is less than or equal to $q$, we have $\tilde{u}_{(k)} \geq \tilde{u}_{(k_0)} \geq u_{(k-q)} \geq u_{(k-b)}$ if $k \geq b+1$. The proof of $\tilde{u}_{(k)} \leq u_{(k+q)} \leq u_{(k+b)}$ for $k \leq m - b$ is similar. □

With the help of Lemma 2, we can prove that $\mathrm{Tmean}_b(\cdot)$ is Byzantine resilient when $b$ is greater or equal to the number of Byzantine nodes. The result is formulated as Theorem 1.

**Theorem 1.** *Let $v_1$, ..., $v_m \in \mathbb{R}^d$ be i.i.d. random vectors such that $v_k \sim G$ with $\mathsf{E}[G] = g$ and $\mathsf{E}[G - g]^2 \leq V$. Let $\tilde{v}_1$, $\tilde{v}_2$, ..., $\tilde{v}_m$ be attacked copies of $v_k$'s, where up to $q$ of them are Byzantine. If $q \leq b < m/2$, then $\mathrm{Tmean}_b(\cdot)$ is $\Delta_1$-Byzantine resilient, where*

$$\Delta_1 = \frac{mV}{(m-b)^2}.$$

*Proof.* See the Appendix. □

The exact time complexity for evaluating $\mathrm{Tmean}_b(\tilde{v}_1, \tilde{v}_2, \ldots, \tilde{v}_m)$ has no closed-form expression in terms of $(m, b, d)$. However, by sorting in each dimension, we can evaluate $\mathrm{Tmean}_b(\tilde{v}_1, \tilde{v}_2, \ldots, \tilde{v}_m)$ using $O(dm \log m)$ operations. The cost is almost linear in practice, and is not much more expensive than evaluating $\mathrm{Mean}(\cdot)$. Hence, $\mathrm{Tmean}(\cdot)$ improves the robustness with only negligible overhead compared to $\mathrm{Mean}(\cdot)$.

## 5 CONVERGENCE ANALYSIS

In this section, we conduct a convergence analysis for SGD using a Byzantine resilient aggregation rule, such as $\mathrm{Krum}(\cdot)$ and $\mathrm{Tmean}(\cdot)$. Since a general convex optimization setting is used, we quote some classic convex analysis theories (see, e.g., (Bubeck, 2014)), as shown in Lemma 3.

**Lemma 3.** *Let $F \colon \mathbb{R}^d \to \mathbb{R}$ be a $\mu$-strongly convex and $L$-smooth function. Then for $x, y \in \mathbb{R}^d$ and $\alpha \in [0, 1]$, one has*

$$\langle \nabla F(x) - \nabla F(y), x - y \rangle \geq \alpha\mu\|x - y\|^2 + \frac{1-\alpha}{L}\|\nabla F(x) - \nabla F(y)\|^2.$$

*Proof.* The $\mu$-strong convexity and $L$-smoothness of $F(x)$ implies

$$\langle \nabla F(x) - \nabla F(y), x - y \rangle \geq \mu\|x - y\|^2,$$

$$\langle \nabla F(x) - \nabla F(y), x - y \rangle \geq \frac{1}{L}\|\nabla F(x) - \nabla F(y)\|^2.$$

The conclusion is a simple convex combination of these inequalities. □

Using Lemma 3 we establish the convergence of SGD as shown in Theorem 2. The expected error bound consists of a linear convergence term similar to that for usual gradient descent methods, and a term caused by randomness. The theorem also suggests theoretically the largest step size: $\gamma = L^{-1}$.

Table 1: Structure of MLP.

| Layer type | Flatten->fc1->relu1->fc2->relu2->fc3->softmax |
|---|---|
| Parameters | Null->#output128->null>#output128->null->#output10->null |
| Previous Layer | data->flatten->fc1->relu1->fc2->relu2->fc3 |

Table 2: Experiment Summary.

| data set | #train | #test | #rounds | $\gamma$ | Batch size | Evaluation metric |
|---|---|---|---|---|---|---|
| MNIST | 60,000 | 10,000 | 600 | 0.1 | 32 | Top-1 accuracy |
| CIFAR10 | 50,000 | 10,000 | 3,000 | $5 \times 10^{-4}$ | 128 | Top-3 accuracy |

**Theorem 2.** *Suppose that the SGD method with* (3) *is adopted to solve problem* (1)*, where* $F\colon \mathbb{R}^d \to \mathbb{R}$ *is a $\mu$-strongly convex and $L$-smooth function. Let* $v_1^{(t)}$, ..., $v_m^{(t)}$ *be local gradients at $t$-th iteration, and* $\tilde{v}_1^{(t)}$, ..., $\tilde{v}_m^{(t)}$ *be the corresponding attacked copies. If the aggregation rule* $\mathrm{Aggr}(\cdot)$ *is $\Delta$-Byzantine resilient, and the step size $\gamma$ satisfies $\gamma \leq L^{-1}$, then*

$$\mathsf{E}\big[\|x^{(t)} - x^*\|\big] \leq \eta^t \|x^{(0)} - x^*\| + \frac{1 + \sqrt{1 - \gamma\mu}}{\mu}\sqrt{\Delta},$$

*where $x^*$ is the exact solution, and*

$$\eta = \sqrt{1 - \gamma\mu} \geq \sqrt{1 - \frac{\mu}{L}}.$$

*Proof.* See the Appendix. ☐

We remark that if we allow a general $\alpha \in [0, 1]$ in the proof of Theorem 2 and adjust the corresponding step size as $\gamma \leq 2(1 - \alpha)L^{-1}$, the decay ratio $\eta$ is then bounded through

$$\eta = \sqrt{1 - 2\alpha\gamma\mu} \geq \sqrt{1 - 4\alpha(1 - \alpha)\frac{\mu}{L}} \geq \sqrt{1 - \frac{\mu}{L}}.$$

The optimal lower bound is achieved when $\alpha = 1/2$ and $\gamma = L^{-1}$.

## 6 EXPERIMENTS

In this section we verify the robustness and convergence of Tmean aggregation rules by experiments. We use a multi-layer perceptron with two hidden layers to handwritten digits classification on the MNIST data set, in $m = 20$ worker processes, we repeat each experiment for ten times and take the average value. Table 1 shows the detailed network structures of the MLP used in our experiments. Then, we conduct recognition on convolutional neural network in the Cifar10 dataset, repeat each experiment for ten times and report the averaged value. In our experiments, we use $\mathrm{Mean}(\cdot)$ *without* Byzantine attack as a reference. We use top-1 or top-3 accuracy on testing sets as the evaluation metrics. Table 2 lists the details of the data set and the default hyper-parameters for the corresponding model. Our experiments are implemented by Python 3.7.4 with packages Tensor-Flow 2.4.1 and NumPy 1.19.2, on a physical node Intel i7-9700 CPU with 32 GB of memory and two NVIDIA GeForce RTX 1080 Ti.

### 6.1 EXPERIMENT SETTING

In our experiment setting, 6 out of the 20 vectors are Byzantine vectors (i.e., $m = 20$, $q = 6$), different values of $b$ will affect different convergence performance. For the MNIST dataset, we take $b = 6$ and $b = 8$ to train the experiment; and we take $b = 8$ on the Cifar10 dataset to conduct recognition task.

### 6.2 GAUSSIAN ATTACK

In Gaussian attack experiment, We use the Gaussian random vector with zero mean and isotopic covariance matrix with standard deviation 200 instead of some correct gradient vectors.

From Table 3 for the MNIST dataset, we find that compared with $\text{Mean}(\cdot)$ without Byzantine, $\text{Tmean}_b(\cdot)$ works well, and is more robust when $b = 8$. However, without trimming the data, $\text{Mean}(\cdot)$ is not robust under Byzantine attack. The method of $\text{Krum}(\cdot)$ is also robust, while it is relatively weaker than $\text{Tmean}(\cdot)$.

For the Cifar10 dataset, we set $b = 8$. From Table 4, the aggregation rule based on the $\text{Mean}(\cdot)$ is not as robust as other aggregation rules, but after trimming the $\text{Mean}(\cdot)$, the output is more robust than before.

Table 3: Accuracy of different aggregations under Gaussian attack in MLP.

| Aggregation rule | Number of iterations | | | | | | | | | | |
|---|---|---|---|---|---|---|---|---|---|---|---|
| | 100 | 150 | 200 | 250 | 300 | 350 | 400 | 450 | 500 | 550 | 600 |
| Mean without Byzantine | 0.87 | 0.88 | 0.89 | 0.89 | 0.90 | 0.90 | 0.91 | 0.91 | 0.92 | 0.92 | 0.93 |
| Krum | 0.71 | 0.82 | 0.85 | 0.86 | 0.87 | 0.87 | 0.88 | 0.89 | 0.88 | 0.89 | 0.89 |
| Mean | 0.63 | 0.62 | 0.62 | 0.63 | 0.64 | 0.64 | 0.64 | 0.65 | 0.65 | 0.65 | 0.66 |
| Tmean($b = 6$) | 0.80 | 0.82 | 0.83 | 0.84 | 0.84 | 0.85 | 0.85 | 0.86 | 0.86 | 0.86 | 0.87 |
| Tmean($b = 8$) | 0.82 | 0.85 | 0.88 | 0.89 | 0.89 | 0.91 | 0.89 | 0.90 | 0.91 | 0.91 | 0.92 |

Table 4: Accuracy of different aggregations under Gaussian attack in Cifar10.

| Aggregation rule | Number of iterations | | | | | |
|---|---|---|---|---|---|---|
| | 500 | 1000 | 1500 | 2000 | 2500 | 3000 |
| Mean without Byzantine | 0.63 | 0.71 | 0.73 | 0.75 | 0.77 | 0.79 |
| Krum | 0.62 | 0.70 | 0.72 | 0.75 | 0.76 | 0.78 |
| Mean | 0.51 | 0.55 | 0.56 | 0.57 | 0.57 | 0.58 |
| Tmean($b = 8$) | 0.62 | 0.71 | 0.73 | 0.74 | 0.76 | 0.77 |

## 6.3 OMNISCIENT ATTACK

Assuming that the attackers (Byzantine nodes) know all the correct gradients information. For each Byzantine gradient vector, all correct gradients of the gradient are replaced by their negative sum. In other words, this attack attempts to make the parameter server go into the opposite direction.

For the MNIST dataset, we see from Table 5 that when the Byzantine node makes the gradient update direction deviate from the maximum actual update direction, $\text{Mean}(\cdot)$ cannot be robust. On the contrary, $\text{Krum}(\cdot)$ is more robust, while $\text{Tmean}(\cdot)$ shows only weak robustness. Because trim is a part of the data intercepted from the mean value for aggregation, it removes the head and tail parts of the entire data set; such that it does not decrease much compared to the overall set mean accuracy when it is attacked by Byzantine.

For the Cifar10 dataset, Table 6 shows that $\text{Krum}(\cdot)$ is more robust than others. Mean-based aggregation behaved badly in data robustness under omniscient attack. If we trim $\text{Mean}(\cdot)$, it can be clearly seen that, the output becomes more robust. However, under this attack, $\text{Tmean}(\cdot)$ is not as robust as $\text{Krum}(\cdot)$.

Table 5: Accuracy of different aggregations under omniscient attack in MLP.

| Aggregation rule | Number of iterations | | | | | | | | | | |
|---|---|---|---|---|---|---|---|---|---|---|---|
| | 100 | 150 | 200 | 250 | 300 | 350 | 400 | 450 | 500 | 550 | 600 |
| Mean without Byzantine | 0.82 | 0.83 | 0.84 | 0.84 | 0.85 | 0.85 | 0.86 | 0.86 | 0.87 | 0.87 | 0.88 |
| Krum | 0.79 | 0.81 | 0.82 | 0.83 | 0.83 | 0.84 | 0.85 | 0.85 | 0.86 | 0.87 | 0.87 |
| Mean | 0.26 | 0.26 | 0.27 | 0.27 | 0.28 | 0.29 | 0.29 | 0.30 | 0.31 | 0.31 | 0.32 |
| Tmean($b = 6$) | 0.23 | 0.28 | 0.32 | 0.38 | 0.47 | 0.55 | 0.61 | 0.66 | 0.70 | 0.73 | 0.75 |
| Tmean($b = 8$) | 0.50 | 0.52 | 0.57 | 0.61 | 0.64 | 0.67 | 0.70 | 0.72 | 0.74 | 0.77 | 0.79 |

Table 6: Accuracy of different aggregations under omniscient attack in Cifar10.

| Aggregation rule | Number of iterations | | | | | |
|---|---|---|---|---|---|---|
| | 500 | 1000 | 1500 | 2000 | 2500 | 3000 |
| Mean without Byzantine | 0.62 | 0.71 | 0.74 | 0.76 | 0.78 | 0.79 |
| Krum | 0.64 | 0.72 | 0.73 | 0.74 | 0.76 | 0.78 |
| Mean | 0.31 | 0.31 | 0.32 | 0.32 | 0.32 | 0.33 |
| Tmean($b = 8$) | 0.42 | 0.43 | 0.45 | 0.45 | 0.46 | 0.48 |

### 6.4 GENERAL ATTACK WITH MULTIPLE SERVERS

We evaluate the robust aggregation rules under a more general and realistic type of attack. It is very popular to partition the parameters into disjoint subsets, and use multiple server nodes to store and aggregate them. We assume that the parameters are evenly partitioned and assigned to the server nodes. The attacker picks one server, and manipulates any floating number by multiplying $-10^{20}$ with probability of $0.05\%$. Because the attacker randomly manipulates the values, with the goal that in some iterations the assumptions/prerequisites of the robust aggregation rules are broken, which crashes the training. Such an attack requires less global information, and can be concentrated on one single server, which makes it more realistic and easier to implement.

For the MNIST dataset, in Table 7 we evaluate the performance of all robust aggregation rules under multiple serves attack. The number of servers is 20. For $\text{Krum}(\cdot)$ and $\text{Mean}(\cdot)$, we set the estimated Byzantine number $q = 6$. We find that $\text{Krum}(\cdot)$ and not passed. The intercepted $\text{Mean}(\cdot)$ aggregation rules cannot be robust. In this case, they cannot converge to the global optimum. On the other hand, for $\text{Tmean}(\cdot)$, different values of b will affect different convergence performance. And if we take $b = 6$, it can also behave robustly.

For the Cifar10 dataset in Table 8, we can find the advantage of the $\text{Tmean}(\cdot)$. Neither $\text{Krum}(\cdot)$ nor $\text{Mean}(\cdot)$ is robust, while $\text{Tmean}(\cdot)$ can make the learning robust.

Table 7: Accuracy of different aggregations under multiple server attack in MLP.

| Aggregation rule | Number of iterations | | | | | | | | | | |
|---|---|---|---|---|---|---|---|---|---|---|---|
| | 100 | 150 | 200 | 250 | 300 | 350 | 400 | 450 | 500 | 550 | 600 |
| Mean without Byzantine | 0.83 | 0.83 | 0.84 | 0.84 | 0.85 | 0.85 | 0.86 | 0.86 | 0.87 | 0.87 | 0.87 |
| Krum | 0.28 | 0.20 | 0.10 | 0.10 | 0.10 | 0.10 | 0.10 | 0.10 | 0.10 | 0.10 | 0.10 |
| Mean | 0.10 | 0.10 | 0.10 | 0.10 | 0.10 | 0.10 | 0.10 | 0.10 | 0.10 | 0.10 | 0.10 |
| $\text{Tmean}(b = 6)$ | 0.80 | 0.81 | 0.81 | 0.82 | 0.82 | 0.83 | 0.83 | 0.84 | 0.84 | 0.84 | 0.85 |
| $\text{Tmean}(b = 8)$ | 0.81 | 0.82 | 0.82 | 0.83 | 0.83 | 0.83 | 0.84 | 0.84 | 0.85 | 0.86 | 0.86 |

Table 8: Accuracy of different aggregations under multiple server attack in Cifar10.

| Aggregation rule | Number of iterations | | | | | |
|---|---|---|---|---|---|---|
| | 500 | 1000 | 1500 | 2000 | 2500 | 3000 |
| Mean without Byzantine | 0.65 | 0.70 | 0.73 | 0.75 | 0.77 | 0.78 |
| Krum | 0.30 | 0.30 | 0.30 | 0.30 | 0.30 | 0.30 |
| Mean | 0.30 | 0.30 | 0.30 | 0.30 | 0.30 | 0.30 |
| $\text{Tmean}(b = 8)$ | 0.64 | 0.69 | 0.73 | 0.74 | 0.76 | 0.77 |

### 6.5 EXPERIMENT CONCLUSION

From the above experiments, we find that it is difficult for $\text{Mean}(\cdot)$ to ensure the robustness of the data when it is attacked by any Byzantine node. When multiple server nodes are used at the same time, $\text{Krum}(\cdot)$ also becomes vulnerable. However, after the data is properly trimmed, $\text{Tmean}(\cdot)$ can maintain data robustness. Under omniscient attacks, $\text{Tmean}(\cdot)$ is not as robust as $\text{Krum}(\cdot)$.But $\text{Tmean}(\cdot)$ still improves a lot compared to $\text{Mean}(\cdot)$ in this case. The data robustness of $\text{Tmean}(\cdot)$ is also related to the value of $b$. We find that when the value of $b$ is closer to $m/2$, it can achieve stronger robustness.

## 7 CONCLUSIONS

We analyzed the aggregation rules of $\text{Tmean}(\cdot)$. We demonstrated that the effectiveness of our approaches can make the FL model robust to Byzantine attacks. We used three different Byzantine node attacks to show that the original data set can be more robustly handled after partial data trimming and averaging operations.

This work focuses on $\text{Tmean}(\cdot)$. In future work we plan to refine the trimming range of $\text{Tmean}(\cdot)$ and prove its convergence in a non-convex environment. At the same time, we will add momentum on the basis of $\text{Tmean}(\cdot)$, or add some constraints to strengthen its robustness.

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

APPENDIX

*Proof of Theorem 1.* We first assume that $v_k$'s, $\tilde{v}_k$'s, and $g$ are all scalars (i.e., $d = 1$), with variance $V = \sigma^2$. Let $v_{(1)} \leq v_{(2)} \leq \cdots \leq v_{(m)}$ and $\tilde{v}_{(1)} \leq \tilde{v}_{(2)} \leq \cdots \leq \tilde{v}_{(m)}$, respectively, be order statistics of $v_1, v_2, \ldots, v_m$ and $\tilde{v}_1, \tilde{v}_2, \ldots, \tilde{v}_m$. Notice that

$$\mathsf{E}\left[\left(\frac{1}{m-2b}\sum_{k=1}^{m-2b}v_{(k)}-g\right)^2\right] = \frac{1}{(m-2b)^2}\mathsf{E}\left[\sum_{k=1}^{m-2b}(v_{(k)}-g)^2\right]$$

$$\leq \frac{1}{(m-2b)^2}\mathsf{E}\left[\sum_{k=1}^{m}(v_{(k)}-g)^2\right]$$

$$= \frac{1}{(m-2b)^2}\mathsf{E}\left[\sum_{k=1}^{m}(v_i-g)^2\right]$$

$$= \frac{m\sigma^2}{(m-2b)^2}.$$

Similarly, we have

$$\mathsf{E}\left[\left(\frac{1}{m-2b}\sum_{k=2b+1}^{m}v_{(k)}-g\right)^2\right] = \frac{m\sigma^2}{(m-2b)^2}.$$

It follows from Lemma 2 that

$$\frac{1}{m-2b}\sum_{k=1}^{m-2b}v_{(k)}-g \leq \mathrm{Tmean}_b(\tilde{v}_1,\tilde{v}_2,\ldots,\tilde{v}_m)-g \leq \frac{1}{m-2b}\sum_{k=2b+1}^{m}v_{(k)}-g.$$

Therefore

$$\mathsf{E}\left[(\mathrm{Tmean}_b(\tilde{v}_1,\tilde{v}_2,\ldots,\tilde{v}_m)-g)^2\right]$$

$$\leq \max\left\{\mathsf{E}\left[\left(\frac{1}{m-2b}\sum_{k=1}^{m-2b}v_{(k)}-g\right)^2\right],\mathsf{E}\left[\left(\frac{1}{m-2b}\sum_{k=2b+1}^{m}v_{(k)}-g\right)^2\right]\right\}$$

$$\leq \frac{m\sigma^2}{(m-2b)^2}.$$

Now we consider the general case—$v_k$'s, $\tilde{v}_k$'s, and $g$ are $d$-vectors, with variance $V = \sum_{i=1}^{d}\sigma_i^2$, where

$$\mathsf{E}\left[\langle G-g,e_i\rangle^2\right] \leq \sigma_i^2.$$

Using the result for the scalar case, we have

$$\mathsf{E}\left[\langle \mathrm{Tmean}_b(\tilde{v}_1,\tilde{v}_2,\ldots,\tilde{v}_m)-g,e_i\rangle^2\right] \leq \sum_{i=1}^{d}\frac{m\sigma_i^2}{(m-2b)^2}.$$

Thus we conclude that

$$\mathsf{E}\left[\|\mathrm{Tmean}_b(\tilde{v}_1,\tilde{v}_2,\ldots,\tilde{v}_m)-g\|^2\right] \leq \sum_{i=1}^{d}\frac{m\sigma_i^2}{(m-2b)^2} = \frac{mV}{(m-2b)^2}. \qquad \square$$

*Proof of Theorem 2.* Let $g^{(t)} = \mathrm{Aggr}(\tilde{v}_1^{(t)},\ldots,\tilde{v}_m^{(t)})$. Then

$$\|x^{(t+1)}-x^*\| = \|x^{(t)}-\gamma g^{(t)}-x^*\| \leq \|x^{(t)}-x^*-\gamma\nabla F(x^{(t)})\|+\gamma\|\nabla F(x^{(t)})-g^{(t)}\|. \quad (4)$$

It follows from Lemma 3 that

$$\langle \nabla F(x^{(t)}),x^{(t)}-x^*\rangle \geq \frac{\mu}{2}\|x^{(t)}-x^*\|^2 + \frac{1}{2L}\|\nabla F(x^{(t)})\|^2.$$

Then we obtain

$$\|x^{(t)}-x^*-\gamma\nabla F(x^{(t)})\|^2$$

$$= \|x^{(t)} - x^*\|^2 + \gamma^2\|\nabla F(x^{(t)})\|^2 - 2\gamma\langle x^{(t)} - x^*, \nabla F(x^{(t)})\rangle$$

$$\leq \|x^{(t)} - x^*\|^2 + \gamma^2\|\nabla F(x^{(t)})\|^2 - 2\gamma\Big(\alpha\mu\|x^{(t)} - x^*\|^2 + \frac{1-\alpha}{L}\|\nabla F(x^{(t)})\|^2\Big)$$

$$= (1 - 2\alpha\gamma\mu)\|x^{(t)} - x^*\|^2 + \gamma\Big(\gamma - \frac{2(1-\alpha)}{L}\Big)\|\nabla F(x^{(t)})\|^2.$$

Taking $\alpha = 1/2$ yields

$$\|x^{(t)} - x^* - \gamma\nabla F(x^{(t)})\| \leq \sqrt{1 - \gamma\mu}\,\|x^{(t)} - x^*\| = \eta\|x^{(t)} - x^*\|.$$

The estimate (4) simplifies to

$$\|x^{(t+1)} - x^*\| \leq \eta\|x^{(t)} - x^*\| + \gamma\|\nabla F(x^{(t)}) - g^{(t)}\|.$$

By taking the expectation on both sides, we obtain

$$\mathsf{E}\big[\|x^{t+1} - x^*\|\big] \leq \eta\,\mathsf{E}\big[\|x^{(t)} - x^*\|\big] + \gamma\,\mathsf{E}\big[\|\nabla F(x^{(t)}) - g^{(t)}\|\big]$$

$$\leq \eta\,\mathsf{E}\big[\|x^{(t)} - x^*\|\big] + \gamma\sqrt{\mathsf{E}\big[\|\nabla F(x^{(t)}) - g^{(t)}\|^2\big]}$$

$$\leq \eta\,\mathsf{E}\big[\|x^{(t)} - x^*\|\big] + \gamma\sqrt{\Delta}\,.$$

It can then be easily verified that

$$\mathsf{E}\big[\|x^{(t)} - x^*\|\big] \leq \eta^t\|x^{(0)} - x^*\| + \gamma\sqrt{\Delta}\sum_{i=0}^{t-1}\eta^i$$

$$\leq \eta^t\|x^{(0)} - x^*\| + \gamma\sqrt{\Delta}\sum_{i=0}^{\infty}\eta^i$$

$$= \eta^t\|x^{(0)} - x^*\| + \frac{1 + \sqrt{1 - \gamma\mu}}{\mu}\sqrt{\Delta}.$$

Finally, the decay ratio $\eta$ can be bounded through

$$\eta = \sqrt{1 - \gamma\mu} \geq \sqrt{1 - \frac{\mu}{L}}. \qquad\qquad \square$$

