# OpenReview forum: "FEDERATED LEARNING FRAMEWORK BASED ON TRIMMED MEAN AGGREGATION RULES"
_ICLR.cc/2022/Conference — ICLR 2022 Submitted_

### Official Review · Reviewer_VRXh · 2021-10-22

**Correctness:** 2
**Technical Novelty And Significance:** 1
**Empirical Novelty And Significance:** 1
**Recommendation:** 3
**Confidence:** 4

**Main Review:**


# Major Issues
- The trimmed mean studied in this paper has already been thoroughly analyzed in (Yin et al, 2018). Their analysis is much more general, including strongly convex, non-strongly convex, and non-convex case whereas this paper only prove for strongly convex case. Besides, Yin et al. (2018) also presented a near optimal statistical performance while this paper did not, e.g. theorem 2 does not reflect the actual number of Byzantine worker $q$ influence the rate.

- The studied attack is very weak. The proposed method is not robust against SOTA attacks (Xie et al., 2020; Baruch et al., 2019).

- Lack of baselines. There are abundant aggregation rules for Byzantine resilient learning. Even the simple median based algorithms are not thoroughly compared.

# Obvious mistakes
There are many obvious mistakes in the paper
- Last line of page 1, why is the geometric median a variant of mean-based aggregation rule?
- Page 3, "Based on these attack methods, there exist defense methods, such as robust aggregation, secure aggregation, encryption, etc". The secure aggregation and encryption are not defense to these attacks.
- Theorem 2, $0<L\le \mu$?

# Minor Issues
- The reference style should be improved
    - e.g. paragraph 2 of page 1: "...state machine replication strategy (Alistarh et al. (2018))" should be "...state machine replication strategy (Alistarh et al., 2018)".
- "... on the data model Bottou (2010)" should be "... on the data model (Bottou 2010)"
- "This is the problem not fully addressed in previous works" should be "This problem is not fully addressed in previous works"
- Page 1 paragraph 3, the "single miner attack" and "multiple server attacks" seem not very common, could you provide definition or some explanation?
- Figure 1 (a) "Master serves" => "Master servers", caption "miner serves" => "miner servers".
- Page 3, "Zhao et al. expanded" no year information.
- Page 3, "gradient leakage (Bagdasaryan et al. (2020)", however, the reference is backdoor attack, not the graident leakage attack.
- Page 4, Definition 1 needs reformulated.
- In many occurances, the authors says they trimmed the data while they only trimmed the gradient.

====

# Reference

- Dong Yin, Yudong Chen, Ramchandran Kannan, and Peter Bartlett. Byzantine-robust distributed learning: Towards optimal statistical rates. In International Conference on Machine Learning, pp. 5650–5659. PMLR, 2018.
- Xie C, Koyejo O, Gupta I. Fall of empires: Breaking Byzantine-tolerant SGD by inner product manipulation[C]//Uncertainty in Artificial Intelligence. PMLR, 2020: 261-270.
- Baruch M, Baruch G, Goldberg Y. A little is enough: Circumventing defenses for distributed learning[J]. arXiv preprint arXiv:1902.06156, 2019.

**Summary Of The Paper:**

This paper considers the trimmed mean function as the aggregation rule for the byzantine resilient distributed learning. The authors provide a theoretical convergence for strongly convex objectives. Besides, the authors empirically compare trimmed mean with krum and average.

**Summary Of The Review:**

The theoretical contribution of this work is limited and they have ignored many existing defenses and attacks. Besides this paper contains too many obvious mistakes, typos, and the writing/template clearly does not follow the ICLR guideline.

---

> ### Author Response · Authors · 2021-11-22
> **Federal learning framework based on mean aggregation rules**
>
> Dear Reviewers, Area Chairs and Program Chairs,
>
> We sincerely thank all four reviewers for their thorough and constructive comments.
>
> - We have revised Theorem 2.  Though this theorem is closely related to  [1, Theorem 4], it studies the convergence in a different sense,
>   and  leads to different results.  Theorem 2 does not assume the gradients to be sub-exponential as in [1, Theorem 4].  Moreover, by
>   choosing the  same step size, our convergence rate is better.
> - The influence on the number of Byzantine worker q appears implicitly in the constant Delta in Theorem 2---it affects the accuracy instead
>   of convergence rate.
> - Though we do not pay much attention to the median rule, median is a special case of trimmed mean (when 1 <= m-2b <= 2).
> - Since the geometric median lies in the convex hull of the inputs, it can be viewed as a weighted mean (with non-constant weights).
>   Therefore, we regard it as a mean-based rule.
>  We have fixed the minor issues.  Thanks for careful reading!

---

### Official Review · Reviewer_j3Cv · 2021-11-02

**Correctness:** 3
**Technical Novelty And Significance:** 2
**Empirical Novelty And Significance:** 2
**Recommendation:** 3
**Confidence:** 5

**Main Review:**

The problem of robust aggregation in FL is very relevant, and the solution investigated in this study is meaningful.
My main concern with this paper consists in the lack of originality, as there are already a number of works and frameworks focusing on robust FL aggregation. In particular, Trimmed mean has been already proposed in the previously published work of Yin et al [1]. In that work, FL aggregation based on trimmed mean was already proposed and theoretically investigated, along with other additional aggregation methods. This work is not cited by the authors. In particular, Theorem 2 of this paper seems to be already present in Yin et al.

Moreover, this work is not necessarily focused on FL. The authors assume that clients have the same data distributions and perform $K=1$ SGD at every optimization round. The theoretical setting considered is the one of Distributed Learning and its resilience to Byzantine attacks has already been heavily investigated too.

[1] Yin, Dong, Yudong Chen, Ramchandran Kannan, and Peter Bartlett. "Byzantine-robust distributed learning: Towards optimal statistical rates." In International Conference on Machine Learning, pp. 5650-5659. PMLR, 2018.


**Summary Of The Paper:**

This work analyses the use of trimmed mean as robust aggregation rule in federated learning. The study proposes a convergence analysis of the proposed framework, while the robust aggregation schement is assesed on different benchmarks with respect to standard averaging, and to the previous work of Blanchard et al.

**Summary Of The Review:**

The novelty of the contribution seems questionable.

---

> ### Author Response · Authors · 2021-11-22
> **Federal learning framework based on mean aggregation rules**
>
> Dear Reviewers, Area Chairs and Program Chairs,
>
> We sincerely thank all four reviewers for their thorough and constructive comments.
>
> - Theorem 2 in our paper is closely related to [1, Theorem 4].  However,  these theorems study the convergence in different senses, and lead
>   to different results.  Theorem 2 does not assume the gradients to be sub-exponential as in [1, Theorem 4].  Moreover, by choosing the
>   same step size, our convergence rate is better.

---

### Official Review · Reviewer_AKLN · 2021-11-02

**Correctness:** 3
**Technical Novelty And Significance:** 2
**Empirical Novelty And Significance:** 2
**Recommendation:** 3
**Confidence:** 4

**Main Review:**

This paper is of good readability. The main idea of this work is not difficult to understand. However, there are some concerns as listed below:

1. The proposed aggregation rule Tmean seems identical to 'coordinate-wise trimmed mean' [1]. Also, the theoretical estimation and analysis in the paper seem rougher than that in [1]. Could the authors comment on this and compare their work with [1]?

2. The empirical results in this work are not solid enough. The authors only compare their methods with Krum and mean. There are many other aggregation rules, such as geometric median, centered-clipping [2], and so on. Also, the authors do not evaluate their methods under some advanced attacks [3, 4].

3. The authors propose Tmean for federated learning (FL), where the data of different clients can be heterogeneous. However, their analysis is based on the i.i.d. assumption, which usually does not hold in FL.


[1] Dong Yin, Yudong Chen, Ramchandran Kannan, and Peter Bartlett. Byzantine-robust distributed learning: Towards optimal statistical rates. In International Conference on Machine Learning, pp. 5650–5659. PMLR, 2018.

[2] Sai Praneeth Karimireddy, Li He, and Martin Jaggi. Learning from history for Byzantine robust optimization. In ICML 2021 - 37th International Conference on Machine Learning, 2021.

[3] Cong Xie, Oluwasanmi Koyejo, and Indranil Gupta. Fall of Empires: Breaking Byzantine-tolerant SGD by Inner Product Manipulation. In UAI - Proceedings of The 35th Uncertainty in Artificial Intelligence Conference, 2020.

[4] Moran Baruch, Gilad Baruch, and Yoav Goldberg. A Little Is Enough: Circumventing Defenses For Distributed Learning. NeurIPS, 2019.

**Summary Of The Paper:**

The authors propose the aggregation rule Tmean for federated learning. They show the effectiveness and Byzantine robustness of Tmean by both theoretical analysis and empirical evaluation.

**Summary Of The Review:**

Generally speaking, although most of the results are correct, the main contribution of this work seems very similar to existing works and the empirical evaluation is not solid enough. Due to these reasons, this submission is below the acceptance threshold.

---

> ### Author Response · Authors · 2021-11-22
> **Federal learning framework based on mean aggregation rules**
>
> Dear Reviewers, Area Chairs and Program Chairs,
> We sincerely thank all four reviewers for their thorough and constructive comments.
>
> - Our aggregation rule Tmean is essentially the coordinate-wise trimmed mean in [1], though there are minor differences.  We have added
>   citation for this.  However, our Theorem 2 is not covered by Theorem 4  in [1].  These theorems study the convergence in different senses,
>   and lead to different results.  Theorem 2 does not assume the gradients to be sub-exponential as in [1, Theorem 4].  Moreover, by choosing
>   the same step size, our convergence rate is better.
>
> - The i.i.d. assumption is mainly for theoretical analysis.  The same
>   assumption is used in, e.g., [1].

---

### Decision · Program_Chairs · 2022-01-20

**Decision:**

Reject

**Comment:**

The paper suggests a new aggregation rule for federated learning in order to mitigate Byzantine attacks. However, as the reviewers pointed out, the theoreticial results of the paper are weak and incremental and the experiments are not solid.